# A More Accurate Field-to-Field Method towards the Wind Retrieval of HY-2B Scatterometer

**Xinjie Shi** [1,2], **Boheng Duan** [1] **and Kaijun Ren** [1,2,*]

1   College of Meteorology and Oceanography, National University of Defense Technology, Changsha 410073, China; shixinjie19@nudt.edu.cn (X.S.); bhduan@nudt.edu.cn (B.D.)
2   College of Computer Science and Technology, National University of Defense Technology, Changsha 410073, China
*   Correspondence: renkaijun@nudt.edu.cn; Tel.: +86-138-7480-0930

**Abstract:** In this paper, we present a method for retrieving sea surface wind field (SSWF) from HaiYang-2B (HY-2B) scatterometer data. In contrast to the conventional algorithm, i.e., using a point-to-point (P2P) method based on geophysical model functions (GMF) to retrieve SSWF by spaceborne scatterometer, we introduce a more accurate field-to-field (F2F) retrieval method based on convolutional neural network (CNN). We fully consider the spatial correlation and continuity between adjacent observation points, and input the observation data of continuous wind field within a certain range into the neural network to construct the neural network model, and then synchronously obtain the wind field within the range. The wind field obtained by our retrieval method maintains its continuity and solves the problem of ambiguity removal in traditional wind direction retrieval methods. Comparing the retrieval results with the buoy data, the results show that the root mean square errors (RMSE) of wind direction and wind speed are less than 0.18 rad (10.31°) and 0.75 m/s, respectively. The retrieval accuracy is better than the L2B product of HY-2B.

**Keywords:** spaceborne scatterometer; wind retrieval; F2F; CNN; ambiguity removal

## 1. Introduction

The sea surface wind field is a channel for mass and energy exchange between the atmosphere and the ocean [1], and is an important boundary condition for numerical prediction of ocean environmental parameters [2]. As an important active microwave remote sensing instrument, the spaceborne microwave scatterometer can quickly obtain sea surface wind in all-time and all-weather condition, by observing the sea surface normalized radar cross section (NRCS, $\phi_0$) in multiple azimuths. It is a main satellite remote sensing technique for retrieving sea surface wind field at present [3–6]. On 25 October 2018, the second Chinese ocean power environment satellite (HY-2B) was successfully launched, which is an advanced satellite and has collected a large amount of data for the retrieving the global sea surface wind field.

Currently, wind field retrieval is mainly accomplished by building P2P semi-empirical model functions with statistical methods. P2P retrieval method produces one vector *G* using measurements *S* from one footprint. As early as 1984, Wentz [7] constructed a model function for retrieving the sea surface wind speed (SASS) using data from Ku-band scatterometer, and then demonstrated the mode functions for the National Aeronautics and Space Administration (NASA) Scatterometer (NSCAT). NASA introduced the mode functions for NSCAT and the Quick Scatterometer (QuikSCAT). European Space Agency (ESA) also provided multiple mode functions of the geophysical model functions (GMFs) for C-band radar (CMOD) series. GMF is a widely used method in current retrieval work [8] to describe the quantitative relationship between the observed NRCS and sea surface wind speed, wind direction, etc. by constructing complex model functions [9]. The relationship between GMFs and the physical parameters are nonlinear, moreover, the noise in the

measurement will increase the nonlinear effect. Usually, the satellite observations cannot be directly inverted to retrieve the wind field. In this condition, the maximum likelihood estimation (MLE) method is used for the P2P wind field retrieval. Since the model function has the modulation characteristics of cos(2$\phi$) there are multiple solutions for each group of measured NRCS, leading to the ambiguity, which is commonly dealt with using the vector circle median filter method [10]. To directly solve the ambiguity, Long et al. [11] proposed a field mode model retrieval method (F2F retrieval method), where the NRCS within the whole mowing amplitude is correlated with the wind field model parameters, and the whole wind field is derived simultaneously to ensure the continuity of the wind field, which fundamentally solves the problem of ambiguous solutions removal. However, the mathematical complexity of this method is greatly increased, and the computational cost is also very high, which impairs its applicability.

With the development of machine learning and deep learning techniques, numerical retrieval algorithms based on statistical analysis have become a research hotspot. The application of neural networks provides a new way for establishing the retrieval model of the sea surface wind field, using the powerful learning ability and nonlinear approximation ability of neural networks [12], which can theoretically approximate to a series of complex nonlinear functions. As early as 1993, Thiria et al. [13] applied neural networks to scatterometer wind field retrieval, using a multilayer perceptron network for wind speed retrieval and a neural network classifier for wind direction retrieval. They verified the retrieved results with a simulated wind field. Chen et al. [14] and Kasilingam et al. [15] showed that neural networks perform well in the wind vector retrieval. Lin et al. [16] performed the wind field retrieval based on ERS-2 L2A scatterometer data and studied the retrieval of the sea surface wind field with the new scanning method of scatterometer. The initial neural network approach still poses to ambiguous solutions. To deal with this problem, Cornford et al. [6] proposed a mixture density network (MDN) [17] to retrieve the wind direction, which solved the problem of ambiguous solutions.

In summary, we hope to create a method that can directly solve the problem of ambiguous solutions and can quickly obtain a wind field with higher retrieval accuracy. The ambiguous solutions can be solved using F2F retrieval [11], and the strong complexity of F2F retrieval can be solved using the powerful learning ability of neural networks [12]. The combination of the above two approaches is the core of the proposed method in this paper, which is arranged in the following sections. The datasets used are listed in Section 2. Section 3 describes the dataset matching, retrieval methods and data preparation. The neural network model is constructed and trained in Section 3. We then test the model by comparing the retrieval results of the proposed method with the buoy measurements, and the neural network label values to further validate these retrieval results. These tests and comparisons are presented in Section 4. Finally, the work is discussed and summarized in the last two sections.

## 2. Datasets

### 2.1. HY-2B Scatterometer (HY-2B SCAT) Wind Field Data

The observations are from the microwave scatterometer of the HY-2B satellite, which carries a microwave scatterometer scanning in two beams with pencil-shaped cones. The inner beam is horizontal-horizontal (HH) polarization with an incidence angle of 41.4° and the outer beam is vertical-vertical (VV) polarization with an incidence angle of 48.5°. The specification of HY-2B microwave scatterometer is shown in Table 1.

**Table 1.** Specification of HY-2B microwave scatterometer.

| Technical Parameters | Values |
| --- | --- |
| Working frequency/GHz | 13.256 |
| Observation swath/km | Outer beam: 1700<br>Inner beam: 1350 |
| Ground resolution/km | 25 |
| Backscattering coefficient accuracy/dB | 0.5 |
| Backscattering coefficient range/dB | $-40{\sim}20$ |
| Wind speed range/(m/s) | $2{\sim}24$ |
| Wind speed retrieval accuracy/(m/s) | $\pm2$ (10%) |
| Wind direction retrieval accuracy/(°) | $\pm20$ |

The HY-2B scatterometer data used in this paper are mainly L2A and L2B data. The L2A products are sorted by orbits, with each file containing data from one orbit. The L2A products are organized in terms of wind vector cells, with ranks of 1702 and 810, respectively. L2A products are obtained from L1B data through vector cell matching, and each vector cell has no less than 3 NRCSs with different azimuth/incident angles [18]. In this paper, the L2A product of HY-2B is used as the initial data for our retrieval work.

The organization of L2B level products is similar to that of L2A level products. It stores the unique wind vector solution by removing ambiguous solutions using filtering algorithms. Currently, the National Satellite Ocean Application Service (NSOAS) is releasing two versions of HY-2B SCAT wind products at level 2B using different algorithms, denoted as data processing software (DPS) and Pencil-beam Wind Processor (PWP) in the file names, respectively. In the DPS algorithm, the MLE is used for wind retrieval and the circular median filter method is adopted for ambiguity removal. In contrast, the PWP algorithm uses the multiple solution scheme for wind retrieval and the two-dimensional variational (2DVAR) method for ambiguity removal. In both processors, the NSCAT-4 and the European Centre for Medium-range Weather Forecasts (ECMWF) forecast winds are used as geophysical model function and background winds, respectively. According to the research of Wang et al. [19], the PWP processed wind products are recommended for the scientific community because of better performance. Therefore, we choose the L2B product obtained by the PWP method as the wind field data for our comparison.

*2.2. Advanced Scatterometer (ASCAT) Wind Field Data*

The ASCAT aboard Meteorological Operational Satellite A (MetOp-A) and MetOp-B was launched in 2006 and 2012 by ESA. ASCAT is a C-band scatterometer with three vertically polarized antennas transmitting pulses at 5.255 GHz. There are two observation swaths, both about 500-km wide. Current ASCAT wind products are provided at two spatial resolutions over the global oceans, 12.5 km and 25 km [20]. In this article, the 12.5-km grid level 2 winds from MetOp-A ASCAT and MetOp-B ASCAT are used.

*2.3. Reanalysis of Wind Field Data*

We select the sea surface wind field data at a 10 m height from ERA5, a global atmospheric reanalysis product provided by the European Center for Medium-Range Weather Forecasts (ECMWF), as the label data for our neural network. ECMWF ERA5 is the fifth generation of ECMWF reanalysis data for global climate and weather over the past 40 to 70 years. It covers the global sea surface with a spatial resolution of 0.25° and a temporal resolution of 1 h, and is updated daily. ECMWF ERA5 combines meteorological models and observations, and is the most accurate global sea surface wind field data available.

### *2.4. Buoys and Other Data*

Forty-one NDBC (National Data Buoy Center) buoys with continuous wind vector observation capability more than 50 km offshore are selected for this experiment, mainly located in the coastal waters of the United States. Forty-two TAO (Tropical Atmosphere Ocean) buoys are selected, mainly located in the sea area near the equator of the Pacific Ocean. The surface wind observations with a time resolution of 10 min at a height of 4 or 5 m are selected as the test data for this experiment, and the buoy locations are shown in Figure 1. Sentinel-3A altimeter sea surface wind speed data are also used to cross-validate the wind speed accuracy of the method in this paper.

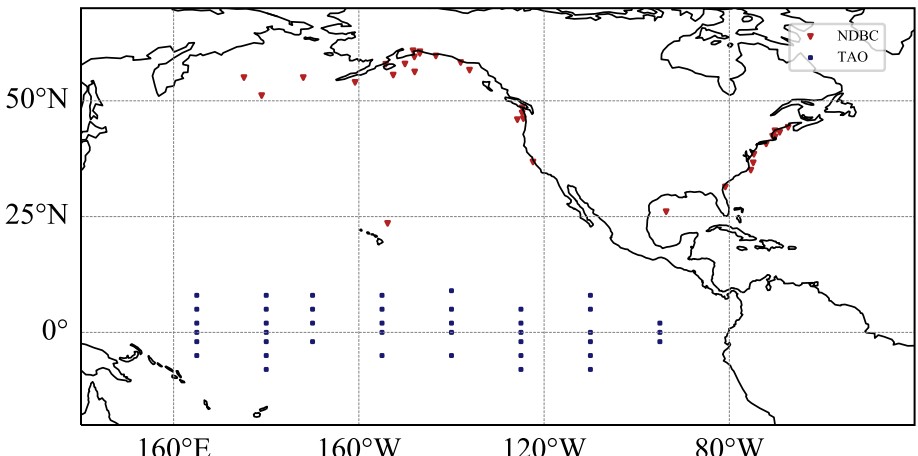

**Figure 1.** Locations of TAO (blue) and NDBC (red) buoys used in this paper.

### 3. Methodology

### *3.1. F2F and CNN*

Most P2P retrieval methods produce ambiguous solutions, requiring a post process to deal with the ambiguity, and such methods are not conducive to maintain the continuity of the wind field [21]. In response to the above problems, we redefine a F2F retrieval method. We construct a continuous wind field from the central sampling point and multiple vector cells in the adjacent specific range, input the NRCS and observed geometric parameters of the continuous wind field into the model, and apply the spatial feature information of the continuous wind field to the retrieval process. Finally, the continuous wind field is calculated at the same time. Figure 2 shows the basic concept of the method.

The size of the continuous wind field has a certain relationship with the retrieval accuracy. The larger the continuous wind field is, the more information it covers, and theoretically the higher the retrieval accuracy will be. However, the increase of the field size will undoubtedly increase the amount of training data. When the field size increases to a certain interval, the cost of computation is not paid off by the accuracy improvement. Therefore, it is crucial to find the most suitable field size. According to Krasnopolsky [21], when the edge length of the wind field increases to 9 (i.e., each continuous wind field is composed of adjacent $9 \times 9$ square area of wind vector cells), the retrieval accuracy is already desirable.

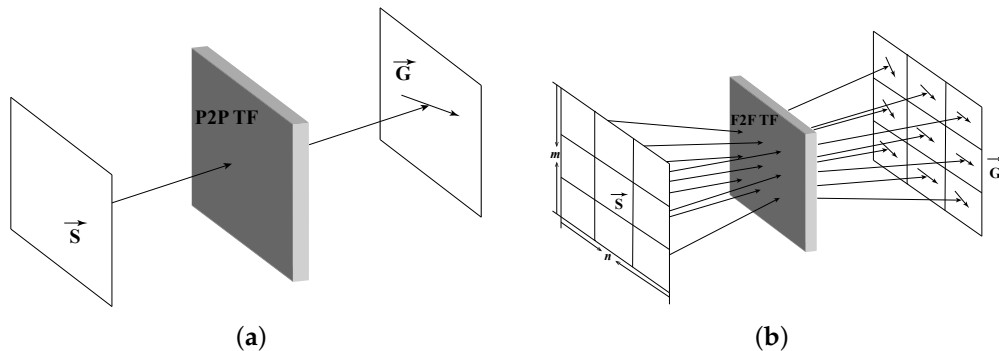

**Figure 2.** Schematic representation of two NN-based approaches: P2P (**a**) and F2F (**b**). What is happening in grey picture? The P2P retrieval method inputs the parameters of a single vector cell (NRCS, etc.) into the P2P TF to obtain the corresponding single wind field (wind speed and direction). The results obtained by the P2P method lack continuity and there are ambiguous solutions. The F2F method inputs the parameters of multiple wind vector cells (m × n in (**b**)) within a certain range into the F2F TF, and retrieve m × n wind fields at the same time. The F2F TF extracts the spatial continuity characteristics of the wind field and applies it to the retrieval process. At the same time, the entire continuous wind field composed of multiple wind vector cells is obtained, which can fundamentally eliminate the ambiguous solutions, and the wind vector cells of the obtained wind field are smoother and more continuous. The F2F TF is a retrieval model, which contains mathematical formulas for the process (retrieval) from the measured parameter data of the scatterometer to the wind field data. The retrieval process is completed by a neural network, and the continuity characteristics of the wind field will be applied to this process. TF represents the transfer function. m × n is the base size, i.e., the number of cells that serve as NN inputs and/or outputs. The case for m = n = 3 is shown.

The F2F method involves complex models, especially when dealing with a large amount of data. Therefore, a method that has a strong ability to learn complex models to complete this task. CNN makes the network easy to optimize, reduces the complexity of the model, and reduces the risk of overfitting. In particular, the continuous wind fields of a specific range can be considered to be images [22], which can be a direct input of the network, avoiding the complicated process of feature extraction and data reconstruction in traditional recognition algorithms [23]. CNN, with these features and advantages, can be a perfect match for our needs.

The F2F method is used to achieve the wind field retrieval with the assist of CNN, thus termed F2F-CNN. The F2F-CNN method needs only an input of the continuous wind field during the network training process, involving no post process for dealing with the ambiguity, and can derive a smooth and continuous wind field.

*3.2. Data Matching*

To make the comparison result more convincing and more authentic, we control the quality of the scatterometer data. We need to eliminate abnormal data values and low-reliability data for different scatterometer products. We only keep the wind vector cells with the quality mark of "Reserved" and the mark value of 0 in the HY-2B scatterometer product as our experimental objects ("Good" in ASCAT, and the mark value is also zero).

According to the requirements of the F2F-CNN method, we divided the obtained data into training data and test data. The multiple orbital L2A data of the microwave scatterometer in May and June 2019 are selected as the training data and the first set of test data. For the second set of test data we select the L2A data observed and generated by the HY-2B microwave scatterometer in May 2020. The second set of test data contain cyclone structures that help demonstrate the reconstruction capability of this retrieval method for cyclones. In addition, we also prepared HY-2B L2B data, ASCAT L2 data and buoy data to validate the F2F-CNN results.

The L2A data of HY-2B stores observations from multiple sampling points in each wind vector cell, which is to ensure that each wind vector cell has no less than three sets

of NRCS with different azimuth/incidence angles and imaging geometry parameters, so that the unique wind speed and direction can be accurately inversed [18]. According to the organization form of ECMWF label data, we re-match the L2A data with the label data wind vector cell to ensure that each label data cell has four sets of NRCS and observation geometric parameters with different azimuth/incident angles. According to the range of each cell of the ECMWF grid, we re-match each sampling point of the L2A data to the ECMWF cell. Subsequently, we classified the data of multiple sampling points in each cell, and classified the sampling points with the same incident angle and azimuth angle into one category (although the polarization method is the same, the incident angle and azimuth angle are also different, therefore, we should set the range, consider that the sampling points with the incident angle and the azimuth angle within a certain range of difference are regarded as the same type of observation point), and finally divided the matching sampling points in each cell into four categories. Finally, we used inverse distance weight (IDW) to interpolate the sampling point data of each type of face element to the center position of the vector cell. This method is implemented by Equations (1) and (2),

$$z_0 = \sum_{i=1}^{n} \frac{1}{D_i^p} z_i \left( \sum_{i=1}^{n} \frac{1}{D_i^p} \right)^{-1}, \tag{1}$$

$$D_i = \sqrt{(x_0 + x_i)^2 + (y_0 + y_i)^2}, \tag{2}$$

where $z_0$ represents the estimated value of interpolation, $z_i$ represents the attribute value of the $i$-th sample ($i = 1, 2, 3, \ldots, n$). $p$ is the power of the distance, and the default is 2, $D_i$ is the distance.

We matched the three closest data in the ECMWF data according to the collection time of each sampling point. According to Equations (3) and (4), the longitudinal and latitudinal component winds in these three ECMWF data are converted into wind speed and wind direction, where the wind direction is saved according to a 0° direction due north and increasing clockwise.

$$spe(u, v) = \sqrt{u^2 + v^2}, \tag{3}$$

$$dir(u, v) = \begin{cases} \arctan(u/v), & v > 0, u \geq 0, \\ \arctan(u/v) + 2\pi, & v > 0, u < 0, \\ \arctan(u/v) + \pi, & v < 0, \\ \pi/2, & v = 0, u \geq 0, \\ 3\pi/2, & v = 0, u < 0, \end{cases} \tag{4}$$

where $u$ is the latitudinal wind component, $v$ is the longitudinal wind component, and $spe$ is the wind speed (in m/s), and $dir$ is the wind direction (in radians). Assuming that the three moments to match the ECMWF are $t_1$, $t_2$ and $t_3$, the corresponding wind speed (wind direction) values are $val_1$, $val_2$ and $val_3$. We obtained the wind speed (or wind direction) value $val$ at time $t$ according to Equations (5) and (6),

$$t_{prime} = \frac{t - t_2}{t_3 - t_2}, \tag{5}$$

$$val = \left( \frac{val_3 + val_1 - 2val_2}{2} + \frac{val_3 - val_2}{2} \right) t_{prime} + val_2, \tag{6}$$

where $t_1 < t_2 < t < t_3$.

For matching HY-2B L2B data and ASCAT L2 data with ECMWF data, we choose the bilinear interpolation method, which is a linear interpolation in two directions perpendicular to each other according to Equations (9)–(11). Suppose we want to obtain the unknown function $f$ at the point $P = (x, y)$ and it is known that the value of the function $f$ at the four points $Q_{11} = (x_1, y_1)$, $Q_{12} = (x_1, y_2)$, $Q_{21} = (x_2, y_1)$ and $Q_{22} = (x_2, y_2)$. Linear

interpolation is first performed in the x-direction based on Equations (7) and (8), and then again in the y-direction based on Equation (9),

$$f(x, y_1) \approx \frac{x_2 - x}{x_2 - x_1} f(Q_{11}) + \frac{x - x_1}{x_2 - x_1} f(Q_{21}), \tag{7}$$

$$f(x, y_2) \approx \frac{x_2 - x}{x_2 - x_1} f(Q_{12}) + \frac{x - x_1}{x_2 - x_1} f(Q_{22}), \tag{8}$$

$$f(P) \approx \frac{y_2 - y}{y_2 - y_1} f(x, y_1) + \frac{y - y_1}{y_2 - y_1} f(x, y_2). \tag{9}$$

The size of the continuous wind field is set to 9 × 9. Each continuous wind field is composed of 81 adjacent wind vector cells with a size of 25 km × 5 km. We traversed each of the obtained data and used each vector cell in turn as the central cell, spreading the vector cell outward in 4 layers to form (1 + 2 × 4) × (1 + 2 × 4) = 81 cells. If all 81 vector cells have the complete data required for the experiment, the 9 × 9 continuous wind field is considered to be a complete training (test) wind field data. Screening the wind vector cells in each direction from 0 to 360° (Figure 3a) and each velocity level from 0 to 25 m/s (Figure 3b) in a balanced manner, we thus obtained 156,815 continuous wind field data for the training and test of the neural network.

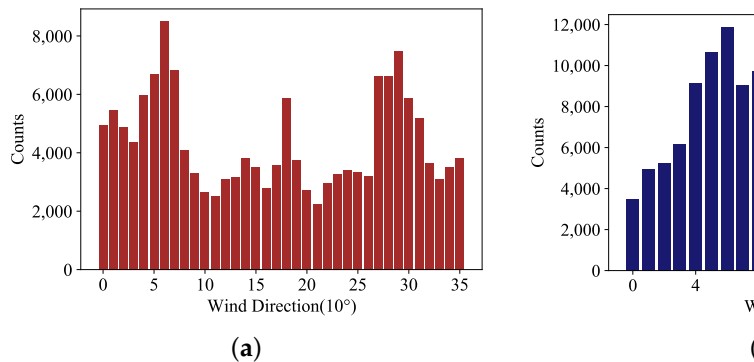

(**a**)  (**b**)

**Figure 3.** Wind direction statistics (**a**) and wind speed statistics (**b**). In (**a**), the interval of the abscissa of the wind direction is 10°, starting from 0, every 10° is divided into one category for statistics. In (**b**), starting from 0, every 1 m/s is counted as a category.

Since the ECMWF data, ASCAT L2 data and the L2B data of HY-2B are both sea surface wind information at a height of 10 m. We need to convert the wind speed of the buoy data to the wind speed at a height of 10 m according to the Equation (10) [24], and the wind direction is not needed to convert,

$$u_{10} = \frac{8.87403 u_z}{\ln(z/0.0016)}, \tag{10}$$

where $z$ represents the height from the sea surface, $u_{10}$ and $u_z$ represent the wind speed at 10 m height and the wind speed at $z$ height, respectively.

Four statistical parameters are used to evaluate the accuracy of wind field data obtained by the various methods mentioned in this paper. These comprised the *RMSE* (Equation (11)), *Bias* (Equation (12)), correlation coefficient (r) (Equation (13))and scatter index (SI) (Equation (14)),which can be expressed as follows:

$$RMSE = \sqrt{\frac{\sum_{i=1}^{n}(S_i - T_i)^2}{n}}, \tag{11}$$

$$Bias = \frac{\sum_{i=1}^{n}(S_i - T_i)}{n}, \tag{12}$$

$$r = \frac{\sum_{i=1}^{n}[(S_i - \overline{S})(T_i - \overline{T})]}{\sqrt{\sum_{i=1}^{n}(S_i - \overline{S})^2 \sum_{i=1}^{n}(T_i - \overline{T})^2}}, \quad (13)$$

$$SI = \frac{\sqrt{(1/n)\sum_{i=1}^{n}[(S_i - \overline{S}) - (T_i - \overline{T})]^2}}{\overline{T}}, \quad (14)$$

where $S$ represents the value to be tested, $T$ represents the true value used for measurement, $\overline{S}$ represents the mean value of the test value, $\overline{T}$ represents the mean value of the true value, and $n$ represents the number of test values.

### 3.3. Establishing and Training the Neural Network

We applied CNN to retrieve the sea surface wind speed and direction. The basic convolutional neural network consists of three structures, i.e., convolution, activation and pooling. The most important process of the entire network is to adjust the network weights through training, the specific structure is shown in Figure 4.

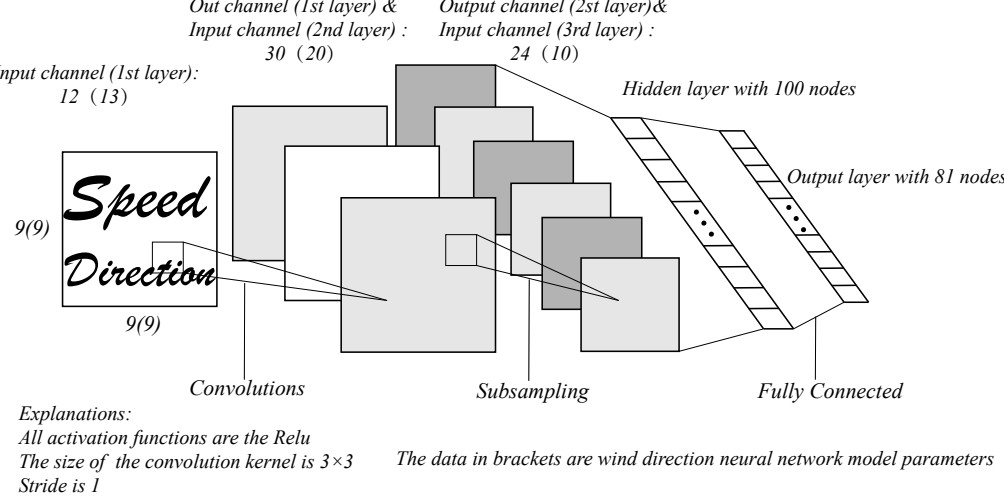

**Figure 4.** The schematic diagram of our CNN. The structure and parameters of the wind speed neural network are different from the wind direction neural network. The structure and parameter settings of the two neural networks are shown in the figure (The data in the brackets are the structural parameters of the wind direction neural network).

The wind speed retrieval is not only related to the NRCS, but also to the incident angle of the antenna beam and the observed azimuth angle [25]. Each training sample contains four different sets of incidence angle, azimuth angle, and NRCS, so the number of input channels of the convolutional layer of the neural network should be 12. To ensure that the whole wind field is derived simultaneously, the node number of the fully connected output layer should be 81, where the output nodes correspond strictly to the positions of the continuous wind field. The 81 output nodes are assumed to be $a_0, a_1, a_2, a_3, \ldots, a_{80}$, the values of the $9 \times 9$ vector cells are $b_{00}, b_{01}, b_{02}, \ldots, b_{08}, b_{10}, b_{11}, \ldots, b_{88}$, the correspondence can be obtained according to Equation (15),

$$b_{ij} = a_{9 \times i + j}. \quad (15)$$

The neural network for wind speed retrieval consists of two convolutional layers and two fully connected layers. The number of input channels and output channels of the first convolutional layer are 12 and 30, respectively. The 30 channels are input to the second layer, which outputs 24 channels. The fully connected network contains a hidden layer with 100 nodes, and there are 81 nodes of the fully connected layer. All activation functions are the ReLU function, the size of the convolution kernel of the two convolution layers is

$3 \times 3$, and the stride is 1. To train the neural network adequately, we set epoch to 100, the batch size to 64, and the loss function to RMSE.

The wind direction retrieval requires not only the 12 parameters used in the wind speed retrieval, but also the wind speed [25]. Therefore, the number of input channels of the wind direction retrieval is 13, and the number of output channels in the first and second convolutional layers are 20 and 10, respectively. The rest of the settings are consistent with the wind speed retrieval model. The loss function of the wind retrieval neural network is still chosen as RMSE, but the calculation method is different. The wind direction error can be calculated in two different directions along the circumference of the circle. Combining with the actual situation, we take the smaller error value of the two directions as our true error based on Equation (16),

$$D_{value}(out, real) = \begin{cases} |out - real|, & |out - real| < \pi, \\ 2\pi - |out - real|, & |out - real| \geq \pi, \end{cases} \quad (16)$$

where $D_{value}$ is the true difference between the neural network output and the direction label, and *out* is the output value of the neural network, and *real* is the label value.

To ensure the generalization ability of the neural network, we randomly disrupted the experimental data. We used about 80% of the obtained data as the training data ($x_{train}$) and the rest as test data ($x_{test}$). We obtained 126,815 training data and 30,000 test data, which are normalized according to Equation (14) in the following order,

$$x_{train} = \frac{x_{train} - x_{mean}}{x_{std}}, \quad (17)$$

where $x_{train}$ is the training data set, and $x_{mean}$ is the mean of the training data, and $x_{std}$ is the standard deviation of the training data.

After 20 iterations of the wind speed retrieval neural network, the loss function value has decreased significantly. After 100 iterations, the RMSE is stabilized around 0.4000 m/s (Figure 5a), and the training of the wind speed retrieval model is completed. The RMSE decreased significantly after 30 iterations, and the error function value (RMSE) is stabilized at about 0.1250 rad after 100 iterations (Figure 5b). The wind direction retrieval model is saved after training.

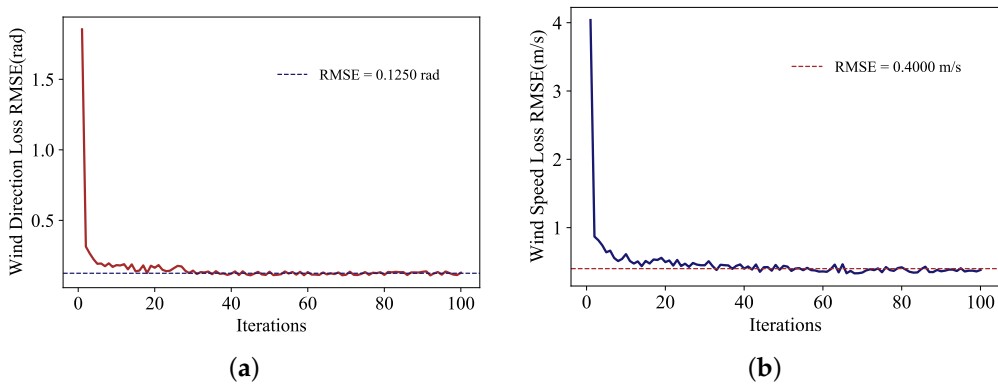

**Figure 5.** Variations of the wind direction RMSE (**a**) and wind speed RMSE (**b**).

## 4. Results and Analysis

### 4.1. Wind Speed Validation Using ECMWF Data for the First Set of Test Data

The first test dataset was used to check the performance of the trained neural network for wind speed retrieval. The RMSE of the wind speed of the test data is 0.4269 m/s. The Bias, SI and r are 0.0528, 0.1415 and 0.9964, respectively. The smaller RMSE, Bias, and SI indicate that the neural network has better predictive capabilities, and the correlation between the two data is extremely high. Figure 6a shows that F2F-CNN has roughly the same response to different wind speed ranges. This method does not reduce the predictive



ability due to excessively high and low wind speeds. On the first test data set, the neural network has better performance for wind speed retrieval. In the process of obtaining L2B products of HY-2B, the ECMWF forecast winds are used as background winds. Therefore, we also compared the HY-2B L2B data with the ECMWF label data (Figure 6b), the RMSE and the correlation coefficient are 1.2042 m/s and 0.9500, respectively. These results show that the F2F-CNN method derives more accurate wind speed than HY-2B L2B product when taking the ECMWF product as the reference.

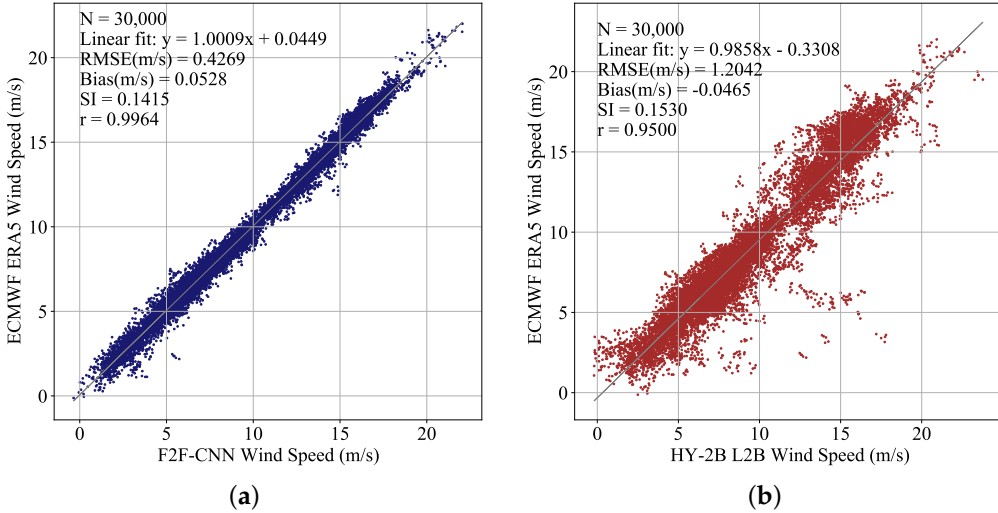

**(a)**                                                                    **(b)**

**Figure 6.** Comparison of label data and F2F-CNN method results (**a**), label data and HY-2B L2B results (**b**). It can be clearly seen that compared to the HY-2B L2B data, the F2F-CNN wind speed is more confirmed to the ECMWF ERA5 wind speed.

Taking the ECMWF wind speed as the reference, we subtracted it from the F2F-CNN wind speed and HY-2B wind speed, respectively. These wind speed difference maps are plotted in Figure 7. Figures in the first row represent a comparison of wind speed above 10 m/s and the second row below 10 m/s. It can be clearly that whether the wind speed range is above 10 m/s or below 10 m/s, the F2F-CNN results are more confirmed to the ECMWF results (Figure 7c,f) when compared to the HY-2B result. When the wind speed is less than 10 m/s, the difference in wind speed between HY-2B and ECMWF can reach 1.8 m/s in some areas (Figure 7c). When the wind speed exceeds 10 m/s, the difference will reach 2.7 m/s (Figure 7f). The conclusion shows that the overall difference between HY-2B L2B wind products and ECMWF data tends to increase with the increase of wind speed. However, the difference between F2F-CNN data and ECMWF data is roughly the same and low in each region (Figure 7b,e).

The ASCAT (MetOp-A and MetOp-B) products are also included in the comparison, and the results are shown in Table 2. Compared with the other three scatterometer wind products, the F2F-CNN data have the smallest RMSE, smallest SI and largest r. Therefore, the F2F-CNN data are closest to the ECMWF product, which is also in line with the experimental expectations. In addition, the HY-2B wind product has the largest error with the ECMWF data, and the wind speed of this product is lower than that of the latter. The two products of ASCAT show relatively similar effects.

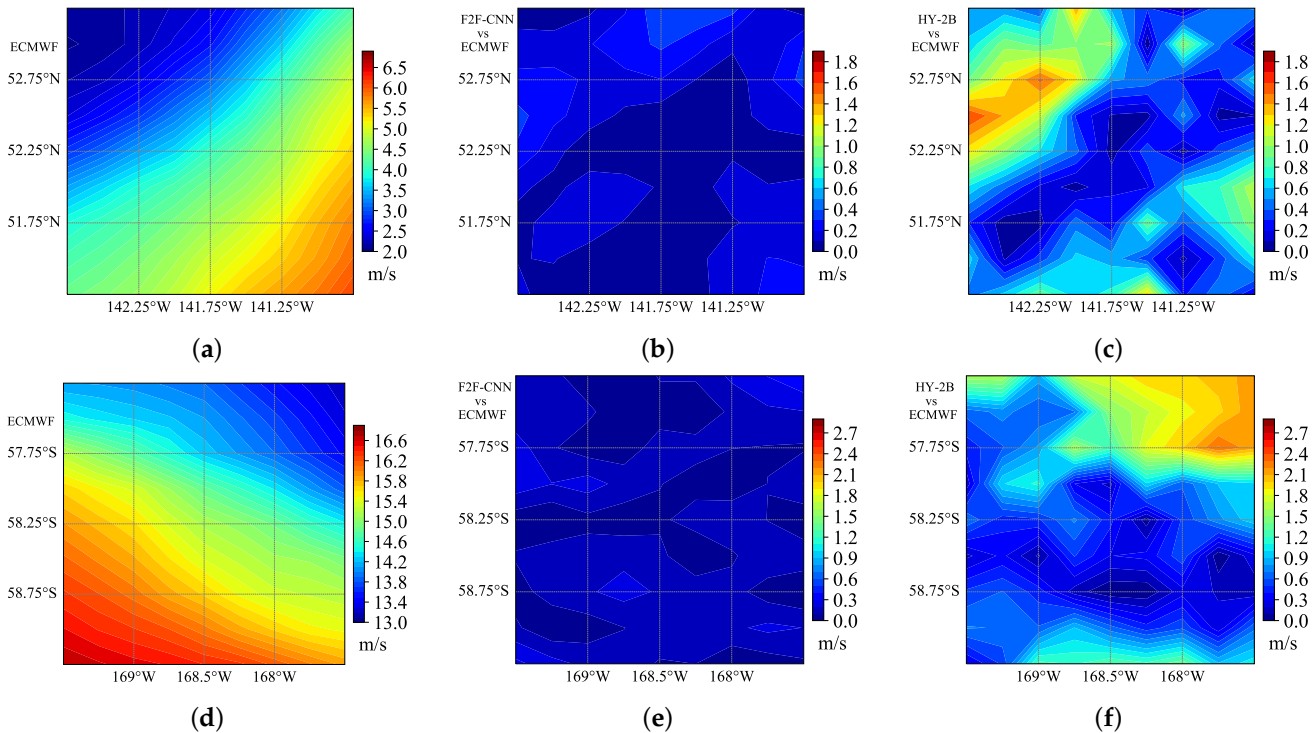

**Figure 7.** (**a**,**d**) show the wind speeds of ECMWF below 10 m/s and above 10 m/s, respectively. (**b**,**e**) show the difference map between F2F-CNN wind speed and ECMWF wind speed. (**c**,**f**) show the difference map between HY-2B wind speed and ECMWF wind.

**Table 2.** Wind speed errors between different wind products and ECMWF data.

| Algorithm | F2F-CNN | HY-2B | MetOp-A | MetOp-B |
|---|---|---|---|---|
| RMSE (m/s) | 0.4269 | 1.2042 | 0.7806 | 0.7428 |
| Bias (m/s) | 0.0528 | −0.0465 | −0.0546 | 0.0843 |
| SI | 0.1415 | 0.1530 | 0.1363 | 0.1537 |
| r | 0.9964 | 0.9500 | 0.9314 | 0.9570 |

A further consistency and stability check of F2F-CNN product is undertaken by cross validating against altimeter data. For this purpose, the calibrated altimeter dataset of Ribal and Young (2019) is used [26]. We compare the MetOp-B L2 data and F2F-CNN result data with Sentinel-3A altimeter data respectively (we have verified above that the accuracy of MetOp-B data are higher than that of HY-2B scatterometer data and MetOp-A data). The RMSE, Bias, SI and r of the F2F-CNN wind speed compared with the altimeter wind speed are 0.6629, −0.0644, 0.1009, 0.9832, respectively. The four data of MetOp-B are 0.7675, −0.1968, 0.0815, 0.9788, respectively [26]. The results show that the F2F-CNN wind speed and MetOp-B wind speed are lower than the altimeter, but both show good agreement. The RMSE and Bias values of the F2F-CNN method are smaller than those of MetOp-B, which means that the accuracy of the F2F-CNN method is superior to that of MetOp-B.

### 4.2. Wind Speed Validation Using Buoy Data for the First Set of Test Data

To further examine the reliability of the model, we compared ECMWF data, F2F-CNN retrieval results, ASCAT L2 data and HY-2B L2B data with buoy data (TAO and NDBC). The RMSE, Bias, SI and correlation coefficient (r) are used for the evaluation, and evaluation results are shown in Tables 3 and 4.

**Table 3.** Wind speed errors between different wind products and NDBC buoy.

| Algorithm | ECMWF | F2F-CNN | HY-2B | MetOp-A | MetOp-B |
|---|---|---|---|---|---|
| RMSE (m/s) | 0.7223 | 0.7137 | 1.5044 | 0.9929 | 0.9585 |
| Bias (m/s) | 0.0601 | 0.0517 | −0.1222 | −0.1102 | 0.1016 |
| SI | 0.1538 | 0.1501 | 0.1504 | 0.1418 | 0.1447 |
| r | 0.9846 | 0.9834 | 0.9395 | 0.9481 | 0.9582 |

**Table 4.** Wind speed errors between different wind products and TAO buoy.

| Algorithm | ECMWF | F2F-CNN | HY-2B | MetOp-A | MetOp-B |
|---|---|---|---|---|---|
| RMSE (m/s) | 0.6779 | 0.7065 | 0.9211 | 0.9061 | 0.7324 |
| Bias (m/s) | −0.3209 | −0.2949 | −0.2163 | 0.2350 | 0.2556 |
| SI | 0.0803 | 0.0860 | 0.1454 | 0.1115 | 0.1272 |
| r | 0.9591 | 0.9558 | 0.8747 | 0.8999 | 0.9566 |

From Table 3, we can see compared with the NDBC buoy data, the wind speed RMSE, Bias and SI of the F2F-CNN data are in the best position among all the five types of data involved in the test. The RMSEs of ECMWF data and F2F-CNN retrieval results are less than 0.75 m/s , which is much better than the accuracy required for operational applications (2 m/s). The accuracy of F2F-CNN retrieval results and ECMWF are close to each other. When compared with TAO buoys (Table 4), the above phenomenon still occurs, and the error values of all five methods have decreased. This is because the TAO buoys are located near the equator, where the geostrophic deflection force is smaller and the wind speed is low and stable, usually lower than 12 m/s. In this condition, the retrieval accuracy is higher than the locations of NDBC buoys with higher wind speed.

### 4.3. Wind Direction Validation Using ECMWF Data for the First Set of Test Data

Similarly, the trained neural network for wind direction retrieval was applied to the first dataset. Figure 8 shows the scatter plot of the F2F-CNN and HY-2B L2B data compared with ECMWF wind directions. The RMSE is 0.1331 rad (about 7.63°) and the correlation coefficient is 0.9982 for the F2F-CNN wind direction. In addition, these two values are 0.2822 rad (16.17°) and 0.9855 for the HY-2B L2B wind direction. The above two indicators show that the F2F-CNN method has an excellent fitting effect on the label wind direction, and the effect is better than HY-2B L2B data.

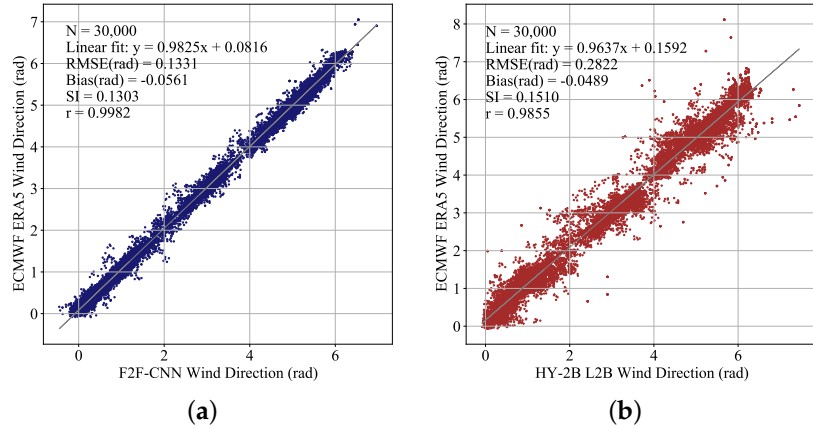

(**a**)　　　　　　　　　(**b**)

**Figure 8.** Comparison of F2F-CNN method results and ECMWF wind direction (**a**), HY-2B L2B results and ECMWF wind direction (**b**). It can be clearly seen that results of the F2F-CNN method are better fitted to the ECMWF wind direction and poses to smaller deviations.

The ECMWF data are used to verify F2F-CNN retrieval results and HY-2B L2B data, to observe the fitting ability of different methods to the wind direction of the wind field under

different wind direction variations, as shown in Figure 9. In the region with insignificant wind direction variation (Figure 9a,b), both F2F-CNN and HY-2B wind directions are very close to the label data. However, in the region with large wind direction variations, the F2F-CNN method fits the ECMWF data well (Figure 9c,e), while the error of HY-2B data becomes significantly larger (Figure 9d,f). Locally, the F2F-CNN method is more sensitive to wind direction variations and can fit well with the wind direction of wind fields with large wind direction variations.

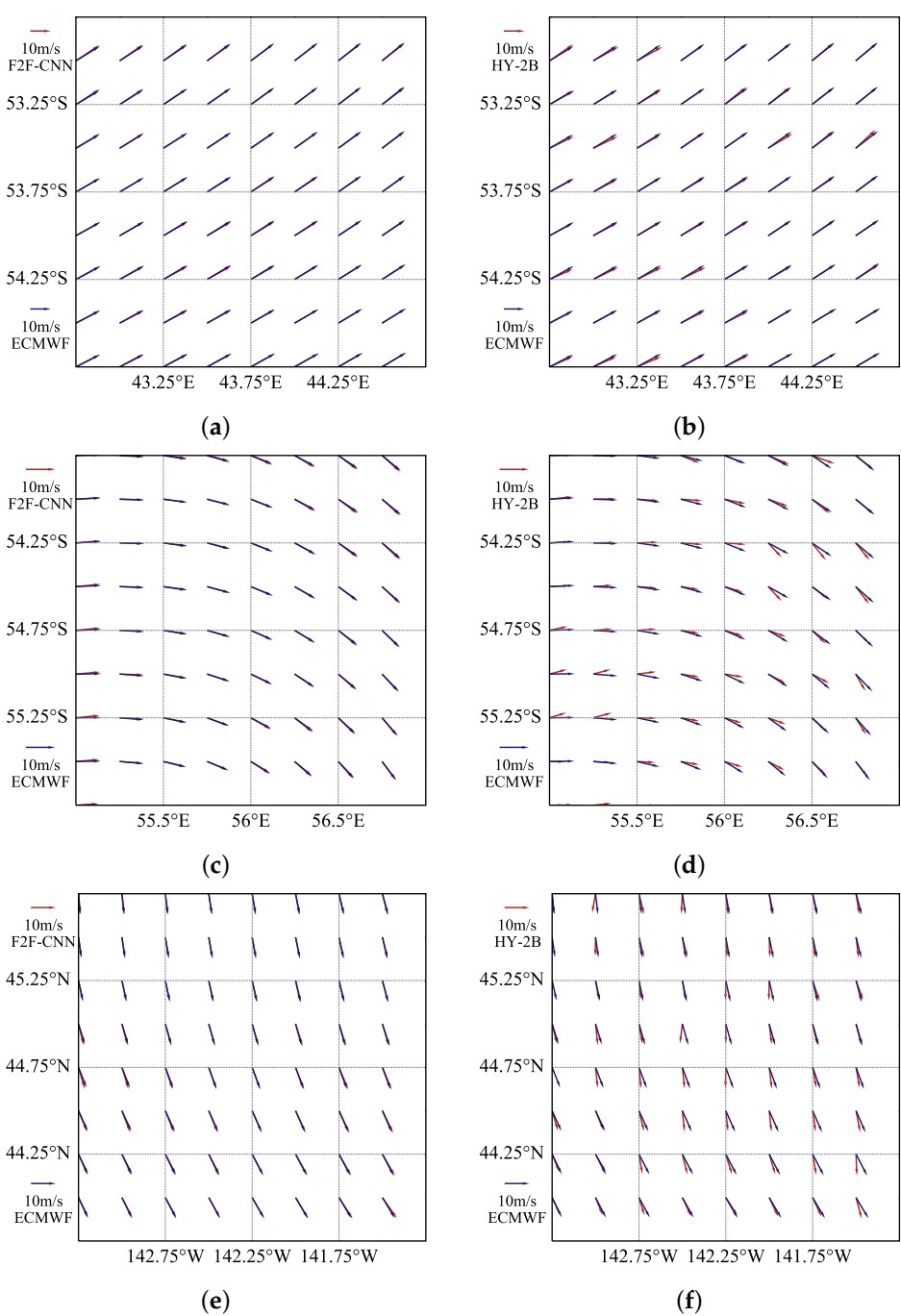

**Figure 9.** (**a,c,e**) show the comparison between the F2F-CNN wind direction and the ECMWF wind direction, and (**b,d,f**) show the comparison between the HY-2B wind direction and the ECMWF wind direction.

To further reflect the advantages of the F2F-CNN results compared with ECMWF data, we also introduce two other scatterometer wind products (MetOp-A and MetOp-B)

for comparison. Table 5 shows that F2F-CNN has the smallest RMSE and the largest r in comparison to the other three scatterometer wind field products. Therefore, the F2F-CNN method is closest to the ECMWF product, which is also in line with the experimental expectations. The error value of MetOp-B data are significantly smaller than that of MetOp-A, i.e., the MetOp-B wind directions are closer to ECMWF than that of MetOp-A.

**Table 5.** Wind direction errors between different wind products and ECMWF data.

| Algorithm | F2F CNN | HY-2B | MetOp-A | MetOp-B |
|-----------|---------|-------|---------|---------|
| RMSE (rad) | 0.1331 | 0.2822 | 0.2629 | 0.2310 |
| Bias (rad) | −0.0561 | −0.0489 | 0.0443 | 0.0288 |
| SI | 0.1303 | 0.1510 | 0.1636 | 0.1489 |
| r | 0.9982 | 0.9855 | 0.9876 | 0.9813 |

*4.4. Wind Direction Validation Using Buoy Data for the First Set of Test Data*

To further verify the accuracy of the model, we introduce the buoy data for comparison. From the comparison results with NDBC buoy in Table 6, we can see that the accuracy of F2F-CNN data are similar to that of the ECMWF data, and is significantly better than other products participating in the comparison. However, the advantages of F2F-CNN become less obvious when compared with TAO buoys (Table 7), and the reasons have been explained above. It is worth mentioning that when compared with TAO buoys, the wind direction Bias value of HY-2B L2B reaches 0.2137, which is significantly higher than the two products of ASCAT. The HY-2B L2B products have relatively large errors in areas with lower wind speeds and smaller wind direction variations. The RMSEs of both ECMWF and F2F-CNN method are less than 0.19 rad (10.89°) and much better than the accuracy required for operational applications (20°).

**Table 6.** Wind direction errors between different wind products and NDBC buoy.

| Algorithm | ECMWF | F2F-CNN | HY-2B | MetOp-A | MetOp-B |
|-----------|-------|---------|-------|---------|---------|
| RMSE (rad) | 0.1819 | 0.1775 | 0.3318 | 0.2476 | 0.2286 |
| Bias (rad) | 0.0140 | 0.0182 | −0.0192 | −0.1375 | −0.0945 |
| SI | 0.0588 | 0.0572 | 0.0838 | 0.0840 | 0.0709 |
| r | 0.9961 | 0.9959 | 0.9898 | 0.8972 | 0.8987 |

**Table 7.** Wind direction errors between different wind products and TAO buoy.

| Algorithm | ECMWF | F2F-CNN | HY-2B | MetOp-A | MetOp-B |
|-----------|-------|---------|-------|---------|---------|
| RMSE (rad) | 0.1698 | 0.1696 | 0.2506 | 0.2188 | 0.1946 |
| Bias (rad) | 0.0172 | 0.0306 | 0.2137 | −0.1218 | −0.0634 |
| SI | 0.0361 | 0.0356 | 0.0268 | 0.0367 | 0.0215 |
| r | 0.9964 | 0.9963 | 0.9915 | 0.9626 | 0.9891 |

*4.5. Test the F2F-CNN in a Cyclone Wind Field*

F2F-CNN has a good performance on the first set of test data. To test the generalization ability of the neural network model, we used the second set of test data to evaluate the F2F-CNN method. To comprehensively analyze the retrieval results, we tried to select a larger area with a large wind speed variation interval and wind direction covering all directions as much as possible for the test. Cyclone structures can meet the above conditions, which are beneficial to test the model. The test data are in the northern Pacific Ocean, west of Canada and the United States, which contains a complete cyclone structure. Since our

preprocessed data are all continuous wind fields with a size of 9 × 9, a single wind field data cannot cover the entire cyclone structure. Therefore, we combine all the vector cells in the area according to their positions to obtain a cyclone wind field with a size of 21 × 21. Analyze the results of this area, and perform visualization processing.

Taking the ECMWF wind speed as the reference, the RMSE of the F2F-CNN method is 0.4606 m/s, the Bias is close to 0, and the result is ideal. In the entire cyclone zone, the difference between the maximum and minimum wind speed is about 18 m/s, and the wind speed varies greatly. We compare the F2F-CNN inversion result with ECMWF wind speed, and take the absolute value of the difference as the error to show the effect of the F2F-CNN method. It can be seen from the Figure 10a that the error of the F2F-CNN results in different wind speed ranges is relatively uniform, and the absolute difference does not exceed 0.8 m/s. The F2F-CNN method is still sensitive to areas with weaker wind speeds in the center of the cyclone, and can respond well to changes in wind speeds in various parts of the cyclone structure. However, the performance of HY-2B L2B data is not satisfactory. HY-2B L2B has a large wind speed error at the center of the cyclone (Figure 10b) and is not sensitive to weak winds.

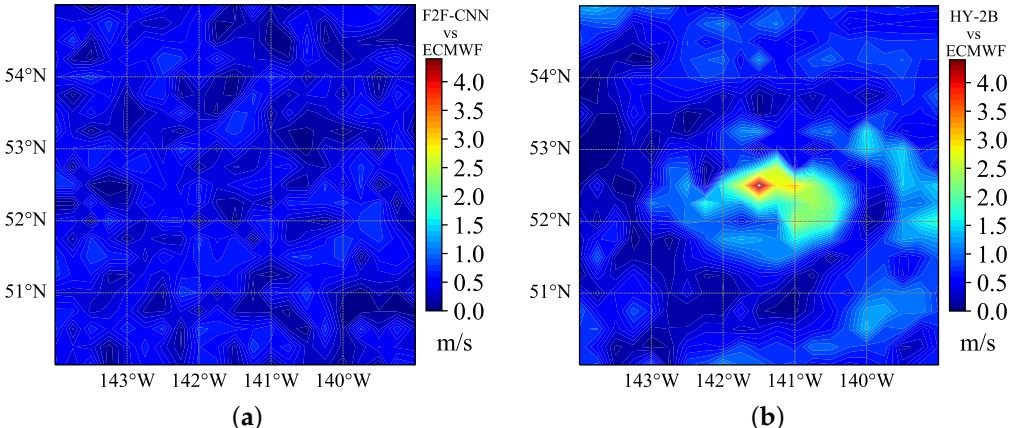

**Figure 10.** (**a**) shows the absolute difference between the F2F-CNN wind speed and the ECMWF data, and (**b**) shows the absolute difference between the HY-2B L2B wind speed and the ECMWF data. It is obvious seen from the figure that the wind speed of the HY-2B L2B data at the center of the typhoon is significantly higher than the ECMWF value, and the overall wind speed error of the F2F-CNN is more uniform and smaller.

NDBC buoy data are introduced to check the accuracy of the speed retrieval results (There is no TAO buoy in the cyclone wind field, so TAO buoy data could not be applied to the comparison experiment), and we compared the F2F-CNN retrieval results, ECMWF data, ASCAT L2 data and HY-2B L2B data with NDBC buoy data, respectively. The comparison results in Table 8 show that the RMSE of F2F-CNN method is less than 0.70 m/s, which is similar to the wind speed accuracy of ECMWF and far better than that of other methods. It is worth noting that the Bias value of this experiment is larger than that of the first test set, because the number of samples involved in the comparison is far less than the previous one, which increases the Contingency. The wind speed of the ECMWF data, the ASCAT data and F2F-CNN data are slightly higher than those of buoy data. However, the HY-2B L2B data are lower, and this phenomenon is more obvious in the center of the cyclone. The HY-2B L2B method is not sensitive to areas with low wind speeds, and its ability to catch weak winds is not strong. Although the HY-2B L2B product has reached the requirements of operational accuracy, it is obviously not as good as the results of ECMWF and F2F-CNN. The test results show that the F2F-CNN has an excellent performance when applied to wind speed retrieval.

**Table 8.** Wind speed errors between different wind products and NDBC buoy in a Cyclone Wind Field.

| Algorithm | ECMWF | F2F-CNN | HY-2B | MetOp-A | MetOp-B |
|-----------|-------|---------|-------|---------|---------|
| RMSE (m/s) | 0.7118 | 0.6961 | 1.5333 | 0.9829 | 0.9585 |
| Bias (m/s) | 0.3841 | 0.3371 | −0.2731 | −0.2422 | 0.1649 |
| SI | 0.1053 | 0.1079 | 0.2140 | 0.1418 | 0.1047 |
| r | 0.9422 | 0.9721 | 0.8903 | 0.8998 | 0.9582 |

The wind direction retrieval using the F2F-CNN was also evaluated in this cyclone field. Compared with ECMWF wind direction, the RMSE of the F2F-CNN wind direction is 0.1105 rad (about 6.33°), and the Bias is close to 0. The comparison (Figure 11b) between the F2F-CNN retrieval results and the ECMWF data shows that the F2F-CNN method can fit the wind in all directions within the entire cyclone ideally. In a single wind vector cell where the wind direction in the middle of the cyclone varies greatly, it is difficult to capture the weak wind. The performance of HY-2B L2B data in this part is unsatisfactory, and the wind direction error is very large, even exceeding 1 rad (Figure 11a). However, the F2F-CNN method can complete this part of the wind field retrieval very well. The predicted value and the real value are almost coincident, and the error is very low. In general, the retrieval results of the F2F-CNN method are very close to ECMWF data.

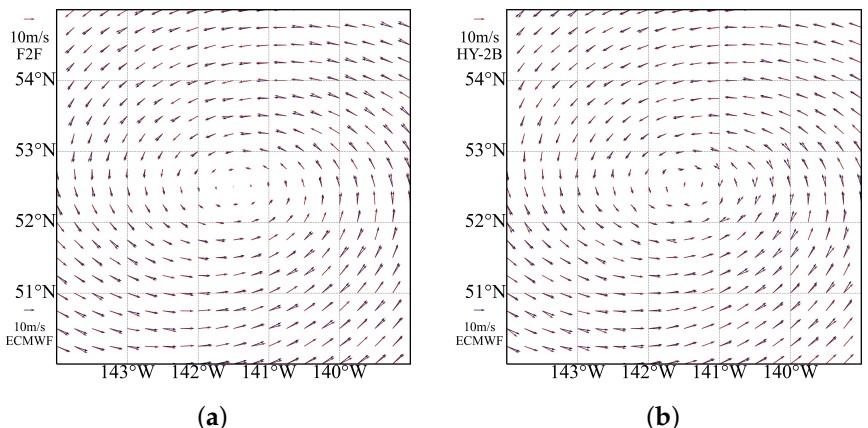

(**a**)         (**b**)

**Figure 11.** The wind direction map derived by F2F-CNN vs. Target (**a**) and HY-2B L2B vs. Target (**b**).

Similarly, the wind direction retrieved by various methods is compared with the NBDC buoy data. Table 9 shows that each method has a better retrieval effect on wind direction. MetOp-B data are better than MetOp-A data in all four indicators, which proves that the performance of MetOp-B has been comprehensively improved compared with MetOp-A. Because the HY-2B L2B data does not have a strong ability to capture the weak wind in the center of the cyclone, it has a large Bias value and RMSE value. The F2F-CNN method is the best among all methods, and the lowest RMSE value and SI value can explain this result.

**Table 9.** Wind direction errors between different wind products and NDBC buoy in a Cyclone Wind Field.

| Algorithm | ECMWF | F2F-CNN | HY-2B | MetOp-A | MetOp-B |
|-----------|-------|---------|-------|---------|---------|
| RMSE (rad) | 0.1841 | 0.1764 | 0.3116 | 0.2868 | 0.2278 |
| Bias (rad) | 0.0710 | 0.1329 | −0.2266 | −0.2510 | −0.1387 |
| SI | 0.0788 | 0.0494 | 0.0967 | 0.0631 | 0.0509 |
| r | 0.9456 | 0.9571 | 0.9102 | 0.9566 | 0.9590 |

Maps of wind speed and wind direction from F2F-CNN, HY-2B and ECMWF are shown in Figure 12. Taking the ECMWF product as a reference, wind speed from the F2F-CNN method is significantly better than that of HY-2B L2B data. In terms of wind speed and wind directions retrieval, the data error of HY-2B L2B is significantly larger, and the center of the cyclone is particularly prominent. HY-2B L2B data are not sensitive to the response of low wind speed areas (Figure 12c), while the F2F-CNN method has a particularly excellent fitting effect. In general, the retrieval result of the F2F-CNN method is very close to ECMWF data, and the wind field is continuous and smooth, indicating that the method in this paper is very successful in the retrieval of the cyclone structure.

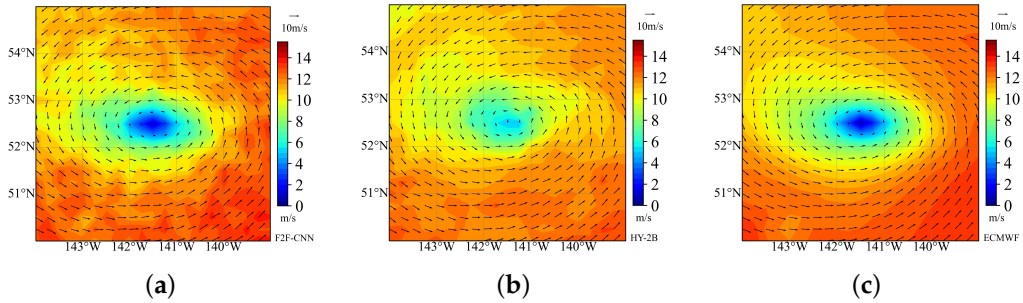

**Figure 12.** The wind speed and wind direction from F2F-CNN (**a**), HY-2B L2B (**b**) and ECMWF (**c**).

*4.6. Summary of Results Analysis*

From the comprehensive performance of the two sets of test data, the RMSE of the F2F-CNN wind speed retrieval is less than 0.75 m/s. This accuracy is very satisfactory. Applying the F2F-CNN to the wind direction retrieval, the RMSE is below 0.18 rad (about 10.31°). The experimental expectation is perfectly met, and the F2F-CNN method can successfully solve the wind field retrieval problem, and the retrieval results have obvious advantages over the HY-2B L2B data and ASCAT L2 data.

## 5. Discussion

In this paper, we present a method for retrieving SSWF from HY-2B scatterometer data. In contrast to the conventional algorithm, i.e., using a P2P method based on GMF to retrieve SSWF by spaceborne scatterometer, we introduce a more accurate F2F retrieval method based on the CNN. The continuity of the wind field is an important part of its spatial characteristics. The P2P retrieval method causes the loss of the connection between adjacent wind vector cells, which makes the wind field isolated. When using P2P methods to retrieve the wind field, there are ambiguous solutions, and the wind field is not smooth enough, so the F2F method is more preferred. In addition, the large amount of sea surface wind field data increases the complexity of the retrieval algorithm. Therefore, it is necessary to use neural network (NN) to solve the problem as it can process big data and learn from them. Based on the F2F and NN, we proposed a F2F-CNN method, which can remove ambiguous solutions and improve the retrieval accuracy.

We choose the L2A product of HY-2B as the basic data we retrieved. In view of the different wind speed and wind direction, we propose two different neural network models. The output of the neural network is the only certain value, so no additional process of removing ambiguous solutions is required. Our neural network can complete the determination of the unique value internally, because we also use the F2F method. The entire wind field composed of 9 × 9 wind vector cells is input into the neural network model, and the convolutional neural network is used to extract the spatial continuity characteristics of each wind field, which is applied to the retrieval and output process of the unique solution.

To fully verify the retrieval accuracy of the F2F-CNN method, we compare the retrieval results of F2F-CNN, HY-2B L2B data, MetOp-A L2 data, and MetOp-B L2 data with ECMWF data. The results show that the F2F-CNN method is closer to ECMWF data than other

data involved in the comparison (This is in line with experimental expectations, because the label data used for neural network training is ECMWF wind data). Especially in the cyclone area where wind speed and direction vary greatly, this method also fits ECMWF data well. The F2F-CNN has a strong ability to capture weak wind, and the fitting effect in different wind field areas is very uniform.

In addition, the buoy data (NDBC and TAO) is used to verify the authenticity of the five retrieved products or results mentioned above. The results show that the retrieval accuracies (RMSE) of wind direction and wind speed with the use of the F2F-CNN is about 10° and 0.75 m/s respectively, largely improved compared with the official products. The HY-2B L2B has no advantage in the five methods involved. This is the reason we proposed the F2F-CNN method, which is to make better use of excellent HY-2B L2A data for retrieval and improve the accuracy of retrieval. The effect of MetOp-B data are better than the MetOp-A data, especially in terms of wind direction. This also verifies that MetOp-B is successful, and its performance is greatly improved compared to MetOp-A. Since the TAO buoy is located near the equator where the deflection force of the earth rotation is small, the sea surface wind is relatively stable. When using TAO buoys to test the five methods, the error value is slightly smaller than when using NDBC buoys (The multiple retrieval methods are more accurate in predicting steady wind). It is very important to accurately retrieve the wind field in areas where the sea surface wind varies drastically, and F2F-CNN method has such a capability.

Combining the comparison results of the F2F-CNN method with the HY-2B L2B and ASCAT L2 data, we can conclude that the retrieval accuracy of F2F-CNN method is better than that of other methods. The F2F-CNN method retrieves the obtained wind field without the need for an additional process of removing ambiguous solutions, and the obtained wind field is smoother and has better continuity.

Apart from the presented work, we have made several additional attempts. We changed the structure of CNN to increase the depth and complexity of the neural network, but this change leads to insignificant improvements of the retrievals. Further innovation of the neural network is a key task. In future work, we plan to retrieve the two-dimensional wind field and to further improve the accuracy of the wind field retrieval.

## 6. Conclusions

Considering the shortcomings and problems of the conventional SSWF retrieval method, this study proposed a F2F method based on the CNN, which has been experimented with the use of microwave scatterometer data obtained by the Chinese HY-2B satellite. This new method takes advantages of the spatial characteristics of the continuous wind field and the fast computational ability of the neural network, and avoids the need for post processing to deal with the ambiguous solutions. ECMWF ERA5 data are used as the label data of F2F-CNN, and HY-2B L2A data are used as the training data of this method. F2F-CNN improves the accuracy of retrieval using HY-2B L2A data. Compared with the buoy observations, the wind speed RMSE of the retrieval result obtained by the F2F-CNN method is less than 0.75 m/s, the wind direction RMSE is less than 0.18 rad (10.31°), and the accuracy is significantly better than the HY-2B L2B products. We also introduced MetOp-A and MetOp-B data to compare with the F2F-CNN method, and F2F-CNN method has better retrieval accuracy than these two methods. In addition, the performance of MetOp-B has obvious advantages over MetOp-A in terms of wind farm retrieval.

In the cyclone wind field area where the wind speed changes greatly, the F2F-CNN method still performs well, and it can fit ideally in all parts of the cyclone. F2F-CNN is sensitive to wind fields with different wind speeds, especially the ability to capture weak winds is better than other methods such as HY-2B L2B data, and the retrieved wind field is very smooth and has strong continuity.

In summary, the F2F-CNN method is successful. This method does not require an additional process of removing the ambiguous solutions, and can also obtain more accurate, smooth and continuous wind field products.

**Author Contributions:** Conceptualization, X.S. and B.D.; Data curation, X.S.; Formal analysis, X.S., B.D. and K.R.; Funding acquisition, K.R.; Investigation, X.S.; Methodology, X.S. and B.D.; Software, X.S.; Supervision, B.D. and K.R.; Validation, X.S.; Visualization, X.S.; Writing—original draft, X.S.; Writing—review and editing, X.S. and B.D. All authors have read and agreed to the published version of the manuscript.

**Funding:** This work was supported by the National Key R&D Program of China (Grant No. 2018YFB0203801), the National Natural Science Foundation of China (Grant Nos. 61572510).

**Institutional Review Board Statement:** Not applicable.

**Informed Consent Statement:** Not applicable.

**Data Availability Statement:** Data is freely available.

**Acknowledgments:** MetOp-A and MetOp-B data were obtained from https://data.eumetsat.int/ (accessed on 20 May 2021). Sentinel-3A data were obtained from https://ladsweb.modaps.eosdis. nasa.gov/ (accessed on 20 May 2021). HY-1C/HY-2B/CFOSAT data were obtained from https: //osdds.nsoas.org.cn (accessed on 15 March 2021)). The authors would like to thank NSOAS for providing the data free of charge. Use of the reference data of ECMWF ERA-5 (https://cds. climate.copernicus.eu/cdsapp#!/home (accessed on 20 March 2021)), NDBC and TAO buoy data (https://tao.ndbc.noaa.gov/ (accessed on 22 March 2021)) is acknowledged. The authors are grateful to ECMWF and NDBC for providing data used in this study.

**Conflicts of Interest:** The authors declare no conflict of interest.

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
