# Peer review of "A More Accurate Field-to-Field Method towards the Wind Retrieval of HY-2B Scatterometer"

_remotesensing, doi:10.3390/rs13122419_

Round 1

Reviewer 1 Report

Report for the manuscript:

A More Accurate Field-to-Field Method towards the Wind Retrieval of HY-2B Scatterometer by Shi et al.

The manuscript reprocesses wind speed obtained from HY-2B scatterometer using Field-to- Field (F2F) Method. The authors claim that the accuracy of their results is better than L2B product of HY-2B.

In my opinion, the manuscript is well written, and it is possible to be published in Remote Sensing after the following issues were addressed.

  1. As above-mentioned, the authors claim that their results are better using F2F method but the method itself is not really clear to me. For example, Figure 2b was not clear. What is happening in grey picture?
  2. Similarly, they mentioned that “filed-to-filed (F2F) retrieval method based on convolutional neural network (CNN)”, However, it is not clear what CNN does? Specifically, as shown in Figure 4, what convolutions are doing including their formulas? How many layers are there? What is calculating in each layer?
  3. Again, since the authors claim that their results are better, I would like to see the comparisons between their results and the wind speed obtained from ASCAT scatterometers, namely METOP-A and METOP-B.
  4. Similarly, it would be good to see also the wind speed comparisons between HY-2B and altimeters, such as Jason-3, Cryosat-2, Sentinel-3A and
  5. I suggest the authors to evaluate some other statistical parameters such as scatter index, bias
  6. As I understand, it is difficult to obtain the raw data from HY-2B. Hence, it will be good if the authors explain the procedure to obtain the
  7. Where the authors data results can be obtained?
  8. Minor issues are as follows:
    • Line 31: QuickSCAT should be QuikSCAT
    • Line 322: What does “?” stands for?
    • Line 419: Date should be Data
    • I think, the following references could be useful to the authors:        https://ieeexplore.ieee.org/abstract/document/8900590             https://ieeexplore.ieee.org/document/8848479

Reviewer 2 Report

The manuscript develops a novel approach based on CNN to retrieve wind vectors from HY-2B scatterometer. I have some concerns:

(1) What is the uncertainty of ERA5 winds used in the paper? Is this uncertainty impact a lot on your CNN model?

(2) The L2B winds product of HY-2B SCAT have two : PWP and DPS, which use the different algorithm, QC flagging, and therefore different performance. So which product of L2B used here? The authors should specify and discuss. 

Wang, H., Zhu, J., Lin, M., Zhang, Y., Chang, Y., 2020. Evaluating Chinese HY-2B HSCAT Ocean Wind Products Using Buoys and Other Scatterometers. IEEE Geoscience and Remote Sensing Letters, 17, 923–927.

J. Zou et al., “The preliminary results of HY-2B microwave scatterom-
eter data,” in Proc. IEEE Int. Geosci. Remote Sens. Symp., vol. 1,
Aug. 2019, pp. 8019–8022.

Reviewer 3 Report

This is an elaborated paper that presents a method for retrieving the sea surface wind field from HY-2B scatterometer data. Results of this method are compared with ECMWF data and HY-2B L2B data are also considered. The analysis includes the wind speed and direction and the RMSE is the main statistic estimator used for comparison. Since the subject, i.e. the obtaining of useful surface meteorological data from observation analysis is vital in regions where surface devices are not available, the paper merits to be published after the introduction of some major changes.

The discussion section must be improved with the introduction of additional references and the comparison between the results of this study and others already published.

The conclusion section could be extended to represent the content of the paper.

Minor remarks.

Line 322. “[?]” a reference number must be included.

Figure 11 caption. What does “visually better” mean?

Reference 9. The year must be included.

Round 2

Reviewer 1 Report

Report for the revised manuscript:

A More Accurate Field-to-Field Method towards the Wind Retrieval of HY-2B Scatterometer by Shi et al.

I have seen that the authors have improved their manuscript in the revised version, and I think, it is now much better and can be published in remote sensing after the following minor revision.

1.It will be good to add scatter index and bias parameters on Figures 6 and 8.

2.It seems to me that the values of statistical parameters for MetOp-A in Table 3 are too high compared to the results in the previous publications. Probably, the authors did not remove the outliers before performing the comparisons. See for examples: https://os.copernicus.org/articles/4/265/2008/https://journals.ametsoc.org/view/journals/atot/37/2/jtech-d-19-0119.1.xml

3.In terms of the comparison with altimeter data, I did not expect the author to compare the wind direction as altimeters do not measure wind direction. The author, however, can compare their wind speed product and wind speed from aforementioned altimeters. Based on this comparison, the reader can see the accuracy of the authors’ wind speed product.

Author Response

Response to Reviewer 1 Comments
Dear reviewer,
Thank you very much for your valuable insights on my paper. I carefully studied the questions you raised and got a lot of inspiration from them. According to your comments, I have answered and explained it point by point. The specific contents are as follows:
Point 1: It will be good to add scatter index and bias parameters on Figures 6 and 8.
Response 1:
I have added scatter index and bias parameters on Figures 6 and 8.
Point 2: It seems to me that the values of statistical parameters for MetOp-A in Table 3 are too high compared to the results in the previous publications. Probably, the authors did not remove the outliers before performing the comparisons.
Response 2:
In order to make the comparison result more convincing and more authentic, we control the quality of the scatterometer data. We eliminate abnormal data values and low-reliability data for different scatterometer products. We only keep the wind vector cells with the quality mark of " Reserved" and the mark value of 0 in the HY-2B scatterometer product as our experimental objects ("Good" in ASCAT, and the mark value is also zero).
Then we recalculated various parameters of each data, and updated the data in Table 3 to Table 8. The results show that our comparison results are very similar to the previous comparison results.
Point 3: In terms of the comparison with altimeter data, I did not expect the author to compare the wind direction as altimeters do not measure wind direction. The author, however, can compare their wind speed product and wind speed from aforementioned altimeters. Based on this comparison, the reader can see the accuracy of the authors’ wind speed product.
Response 3:
Sentinel-3A altimeter sea surface wind speed data is also used to cross-validate the wind speed accuracy of the method in this paper (the last part of section 4.1.).
A further consistency and stability check of F2F-CNN product is undertaken by cross validating against altimeter data. For this purpose, the calibrated altimeter dataset of Ribal and Young (2019) is used [1]. We compare the MetOp-B L2 data and F2F-CNN result data with Sentinel-3A altimeter data respectively (We have verified above that the accuracy of MetOp-B data is higher than that of HY-2B scatterometer data and MetOp-A data). The RMSE, Bias, SI and r of the F2F-CNN wind speed compared with the altimeter wind speed are 0.6629, -0.0644, 0.1009, 0.9832, respectively. The four data of MetOp-B are 0.7675, -0.1968, 0.0815, 0.9788, respectively [1]. The results show that the F2F-CNN wind speed and MetOp-B wind speed are lower than the altimeter, but both show good agreement. The RMSE and Bias values of the F2F-CNN method are smaller than those of MetOp-B, which means that the accuracy of the F2F-CNN method is superior to that of MetOp-B.
[1] Ribal, A.; Young, I.R. Calibration and Cross Validation of Global Ocean Wind Speed Based on Scatterometer Observations. Journal of Atmospheric and Oceanic Technology 2020, 37

Reviewer 2 Report

The authors addressed all my concerns in the revision, improving the manuscript greatly. I think the paper in the present form is suitable for publication in RS.

Author Response

Dear reviewer,

Thank you very much for your Comments and Suggestions.

Reviewer 3 Report

The changes suggested were introduced by the authors.

Author Response

(The authors gave the same response as above.)
